# Pathways to Environmental Inequality: How Urban Traffic Noise Annoyance Varies across Socioeconomic Subgroups

**DOI:** 10.3390/ijerph192214984

**Published:** 2022-11-14

**Authors:** Peter Preisendörfer, Heidi Bruderer Enzler, Andreas Diekmann, Jörg Hartmann, Karin Kurz, Ulf Liebe

**Affiliations:** 1Institute of Sociology, University of Mainz, Jakob-Welder-Weg 12, D-55128 Mainz, Germany; 2School of Social Work, Zurich University of Applied Sciences, Pfingstweidstr. 96, CH-8037 Zurich, Switzerland; 3Environmental Research Group, ETH Zurich, WEP H18, CH-8092 Zurich, Switzerland; 4Institute of Sociology, University of Leipzig, Beethovenstr. 15, D-04107 Leipzig, Germany; 5Research Centre Global Dynamics, University of Leipzig, Strohsackpassage, D-04109 Leipzig, Germany; 6Institute of Sociology, Georg-August-University Göttingen, Platz der Göttinger Sieben 3, D-37073 Göttingen, Germany; 7Department of Sociology, University of Warwick, Coventry CV4 7AL, UK

**Keywords:** noise annoyance, noise exposure, housing attributes, environmental susceptibility, socioeconomic background

## Abstract

The article investigates how socioeconomic background affects noise annoyance caused by residential road traffic in urban areas. It is argued that the effects of socioeconomic variables (migration background, education, and income) on noise annoyance tend to be underestimated because these effects are mainly indirect. We specify three indirect pathways. (1) A “noise exposure path” assumes that less privileged households are exposed to a higher level of noise and therefore experience stronger annoyance. (2) A “housing attributes path” argues that less privileged households can shield themselves less effectively from noise due to unfavorable housing conditions and that this contributes to annoyance. (3) Conversely, an “environmental susceptibility path” proposes that less privileged people are less concerned about the environment and have a lower noise sensitivity, and that this reduces their noise annoyance. Our analyses rest on a study carried out in four European cities (Mainz and Hanover in Germany, Bern and Zurich in Switzerland), and the results support the empirical validity of the three indirect pathways.

## 1. Introduction

Since the emergence of the environmental justice movement in the US in the 1980s, the unequal social distribution of environmental risks has become an important research topic [1,2,3,4]. To investigate environmental inequalities, empirical studies usually concentrate on testing the hypothesis that less privileged population groups (ethnic minorities, low-income households, etc.) are confronted with more serious environmental threats in their everyday life than privileged ones. Whether this so-called social gradient hypothesis of environmental bads holds true, can be examined by referring to objective risk exposure data and/or subjective risk evaluations. Most studies in the tradition of environmental inequality follow an approach that focuses on objective risks. Nevertheless, the joint evaluation of objective and subjective risks is a preferable research strategy because objective and subjective representations may not go hand in hand [5,6]. 

In this article, we consider both objective and subjective data, but focus on subjective risks. The research question is whether and how social background variables affect perceived environmental disamenities. For our empirical analyses, we look at annoyance due to residential road traffic noise as a prominent example of local environmental risks. The selection of noise as environmental threat can be justified both substantively and—in terms of previous research—strategically.

Noise is an important and growing stress factor in urban life. It is well known that long-term noise exposure has negative impacts on people’s subjective well-being and personal health [7,8]. Furthermore, noise is often connected with air pollution and other environmental risks, and is thus a proxy for unfavorable environmental conditions. Compared to other environmental risks, noise can more easily be perceived by individual actors, and therefore seems particularly suitable for an assessment of potential discrepancies between objective conditions and subjective evaluations.

In terms of scientific knowledge and evidence, noise and sound research is a highly developed and sophisticated research field [9,10]. Bearing little relation to research on environmental inequality, noise research is a relatively independent subfield of applied psychology—with a close affiliation to public health. We intend to show in this article that both sides, environmental inequality research on the one hand and noise research on the other, could benefit from being brought into a closer relationship.

Against this background, our article aims in two directions. First, it contributes to the study of noise annoyance within noise and sound research. Our innovative element is that we elaborate—in a more detailed way than previous studies—on the effects of socioeconomic variables and inequality dimensions on annoyance due to urban traffic noise. Second, the article contributes to environmental inequality research: a recourse to the stock of knowledge, theoretical models, and the methodology of noise research could be useful for the study of environmental risks in general, including the unequal social distribution of such risks.

The article is structured as follows. In Section 2, we highlight the theoretical and empirical background and develop our hypotheses. The background section reviews experiences of noise annoyance research, including a focus on the role of socioeconomic variables and environmental inequality. Our hypotheses will differentiate between the direct and indirect effects of socioeconomic variables on noise annoyance. Section 3 introduces our data gathered in four European cities (Mainz and Hanover in Germany, Bern and Zurich in Switzerland). It further describes the dependent and independent variables, and sets out the statistical methods. In the results Section 4, we present our findings concerning the total, direct, and indirect effects of socioeconomic variables on noise annoyance. A discussion and conclusions Section 5 closes the article.

## 2. Background and Hypotheses

### 2.1. Theoretical and Empirical Background

Noise annoyance, i.e., adverse feelings of resentment, anger, discomfort, or offense about ambient noise that interferes with daily activities [11], is a major topic in noise research. Feelings of annoyance serve as the main indicator of the subjective assessment of objective noise exposure conditions. There are standardized methods for measuring noise annoyance in surveys, and relatively well-confirmed empirical models of its most important determinants.

Concerning these determinants, research shows that annoyance depends on both acoustical and non-acoustical factors [12,13,14,15,16,17]. The most important acoustical factor is the objective exposure to noise; so-called dose–response models try to depict its relation to subjective noise annoyance [18,19]. The most prominent non-acoustical factor is noise sensitivity, usually seen as a personality trait. 

In both groups of factors, however, there are additional influences that have been proved or suggested to be significant predictors of noise annoyance. When we are interested in indoor noise annoyance caused by outdoor noise exposure, it seems particularly relevant to take the attributes of the housing into account. For living comfort, subjective well-being, and health effects, indoor rather than outdoor noise is crucial, and there are more or less effective ways of preventing external noise from intruding into the building and thus becoming subjectively annoying. Housing and dwelling attributes, which can be subsumed within the group of acoustical factors [16], include the size of the residence, the sound insulation of the building, the quality of the windows, and the internal functional arrangement of rooms. 

The list of additional non-acoustical factors, most often personal and social attributes, is even longer than that of acoustical ones [12,13,20]. Besides socio-demographic variables (gender, age, etc.), general attitudes, such as environmental concern, and specific attitudes toward the noise source have been shown to be relevant for annoyance responses. Fear of harm connected with the noise source, individual coping capacity, and expectations of the future development of the noise situation are other non-acoustical factors that feature within the debate and corresponding research.

Socioeconomic variables, which belong to this group of non-acoustical factors and constitute the main interest of this article, do not play an important role within noise research. Relatively few empirical studies have focused on socioeconomic inequalities in noise annoyance [6,21,22]. Fyhri and Klaeboe [23] summarize: “within noise research there is rarely any discussion of the relevance of SES [socioeconomic status] for the impacts of noise, nor of the possible mechanisms involved in producing differences of annoyance” (p. 28). This applies not only to SES, but also to income and education. Education, income, and SES are usually grouped under the rubric of social background variables, and reviews of empirical studies regularly conclude that—after controlling for other influence factors—they do not have significant effects on noise annoyance [12,13,16]. This means that regression models, which include a set of proposed determinants of annoyance in a single step, yield insignificant direct effects of social background variables. Nevertheless, some noise researchers, including Fields and Miedema in their review articles, concede that there may be indirect effects, and they recommend more detailed studies of these effects.

A study following this recommendation is Fyhri and Klaeboe [23]. Based on surveys in Norway, the authors investigated the direct and indirect effects of income on urban road traffic noise annoyance. The indirect effect that Fyhri and Klaeboe concentrate on is the mediation via noise exposure. They started with the hypothesis that privileged population groups “buy themselves out of noisy neighbourhoods” (p. 27), i.e., high-income people move out of areas with noise exposure, and this results in their experiencing lower noise annoyance. In accordance with previous research, Fyhri and Klaeboe did not find a direct effect of income on noise annoyance. Contrary to their expectations, however, they also found only partial confirmation for the hypothesis that high-income households have a lower noise exposure. The hypothesis was confirmed in small-to-medium size cities, but not for the large city of Oslo, the capital of Norway. The authors explain their Oslo finding by the fact that Oslo is a highly attractive city for young urban professionals. This group has a preference for living in the capital city, and they trade off urban noise and air pollution against the advantages of living there.

Taking a broader perspective, Fyhri and Klaeboe’s “residential buy-out hypothesis” can be embedded in the field of environmental inequality research. Besides describing environmental social disparities, this research tries to explain what causal mechanisms generate the social gradient of exposure to environmental risks [4,24,25]. When it comes to residential decisions, two mechanisms of selective migration have received most attention in environmental inequality research. (1) The inhabitants of areas with unfavorable environmental conditions may move out when they achieve a higher income. This corresponds to the “residential buy-out hypothesis”. (2) Lower-income groups may settle in areas with unfavorable environmental conditions because rents are lower than in less exposed neighborhoods. Furthermore, minority or migrant groups may experience discrimination in the housing market and may thus be forced into low-quality neighborhoods. This mechanism describes a selective moving-in process. 

Indeed, the majority of environmental inequality studies in the US and European countries report less favorable environmental conditions for ethnic minorities and low-income or low-status groups [1,2,3,21,26,27]. However, there are also remarkable contradictory results in line with Fyhri and Klaeboe’s Oslo finding. In a study of four French cities, Padilla et al. [28] were puzzled by the seemingly paradoxical phenomenon of a positive social gradient in Paris. Exposure to air pollution in the French capital is more severe in city districts with a population of high SES. Pertaining to Rome, Italy, Forastiere et al. [29] also observed a positive association between exposure to traffic-induced air pollution and both income and SES. Rüttenauer [30] explored the association between industrial sites, air pollution, and environmental inequality in German cities. He found that environmental risks for foreigners and migrant workers are higher than for German citizens in most cities. However, there are also cities where this relation is reversed. 

Despite these caveats, research on environmental inequalities suggests for our empirical analyses the indirect two-step path: socioeconomic variables → noise exposure → noise annoyance. We expect that privileged social groups are less exposed to road traffic noise, but this does not necessarily spill over to subjective noise annoyance, i.e., they do not necessarily feel less annoyed. The reason for this potential discrepancy between exposure and annoyance may be one of the two additional indirect pathways proposed in the following section on our hypotheses.

### 2.2. Hypotheses: The Direct and Indirect Effects of Socioeconomic Variables

Since it is relevant for our hypotheses, we begin this section with a specification of our “socioeconomic-group variables”. We use three measures to capture the social background and socioeconomic resources of our respondents: migration background, education, and income. Migration background denotes whether the respondent or at least one of his/her parents was born abroad. Persons with a migration background often have several disadvantages in a new host country—devalued educational credentials, lower skills in the new language, a restricted social network, and direct forms of discrimination. Education aims at a person’s labor market as well as sociocultural resources. Income can be seen as the most direct measure of economic resources and captures the respondent’s financial constraints and opportunities. 

Figure 1 can serve as a road map for our theoretical argumentation and hypotheses. Taken together, we will test four broad hypotheses. The first concerns the direct effects of socioeconomic variables on annoyance from urban road traffic, while hypotheses 2–4 specify indirect, i.e., mediating, effects.

Based on the results of previous research and the theoretical arguments presented above, we predict that—after controlling for the other influencing factors in Figure 1 and a set of additional control variables (not shown in Figure 1)—there are no direct effects of migration background, education, and income on annoyance due to road traffic noise. If we unexpectedly observe significant direct effects, we predict that the strength of these effects is weak, much weaker than the effect of other well-known predictors of noise annoyance (noise exposure, noise sensitivity, etc.).

The preceding Section 2.1 also introduced the indirect “noise exposure path” (path 1 in Figure 1). Research on environmental inequality postulates that processes of selective migration lead to lower levels of noise exposure for persons with higher socioeconomic resources; and from research on noise annoyance, we know (as mentioned above) that noise exposure is the most important predictor of noise annoyance. We expect that migration background yields a significant positive effect on noise exposure, while income yields a significant negative effect. Education usually correlates with income, and our expectation is that the income effect dominates the education effect in the context of noise exposure.

A second indirect path (path 2 in Figure 1) should run through housing attributes. In Section 2.1, we mentioned that housing attributes are important factors influencing indoor noise annoyance and that there are several ways to prevent external noise from intruding into the building. It seems reasonable to assume that privileged households live in dwellings that shield more effectively against residential noise. Diekmann et al. [31] developed this “environmental shielding hypothesis” in more detail. The hypothesis states that better-off social groups reside in dwellings that tend to serve as “structural coping devices” against outdoor noise. Their dwellings are larger, and have more rooms and better noise protection appliances. In our analyses, we will specifically look at four housing attributes: dwelling size; whether or not the bedroom faces the street; window quality; and whether the dwelling has an outdoor garden. Our prediction is that—in a first step—these four attributes will be significantly influenced by socioeconomic variables, and—in a second step—they will have significant effects on noise annoyance. Specifically, we expect that households with a higher income will live in bigger dwellings, less often have a bedroom facing the street, more often have a residence with high-quality soundproofed windows, and more often enjoy an outdoor garden. The reverse should be true for respondents with a migration background, because independent of financial constraints ethnic discrimination is a persistent feature of the housing market in many countries [32]. Contrary to this, when we control for income, additional effects of education on the housing attributes would be surprising. Having a bigger residence (with several rooms) usually means that household members have better opportunities to arrange daily activities in a noise-evading way, and this should reduce noise annoyance. We further assume that those who have a bedroom facing the street are more often annoyed by road traffic noise. On the other hand, both soundproofed windows and an outdoor garden that most often is backyard can be expected to contribute to lower annoyance.

In addition to the “noise exposure path” and the “housing attributes path,” we assume a third indirect path—an “environmental susceptibility path” (path 3 in Figure 1). This path proposes that privileged people are environmentally more concerned and have a higher general noise sensitivity, and this tends to lead to a higher level of noise annoyance. As has been described in Section 2.1, there is strong evidence that noise sensitivity affects noise annoyance, and some evidence that environmental attitudes have a positive influence. The increasing public attention to environmental protection in recent years has contributed to a framing that subsumes noise issues under the broader umbrella of environmental problems [22,33]. This justifies the inclusion of environmental concern in our study. Looking at the socioeconomic variables, there is good reason to assume that educational achievements, rather than migration background and income, mainly matter in this context. Education is usually connected with a higher level of environmental knowledge and knowledge about the negative health effects of environmental bads, and this should stimulate “environmental susceptibility” (for this concept, e.g., [34,35]). Numerous studies in environmental social sciences confirm that education is a robust predictor of environmental concern [36,37]. The evidence concerning the effect of education on noise annoyance, however, is mixed [21,38,39,40].

Whereas the “noise exposure path” and the “housing attributes path” suggest that privileged groups will be less annoyed by road traffic noise, the “environmental susceptibility path” runs in the opposite direction: it suggests that privileged people complain more about noise. These contradictory indirect effects may be one of the reasons why most previous studies did not find that socioeconomic background variables have significant effects on noise annoyance.

## 3. Data, Variables, and Methods

### 3.1. Empirical Data

The main data for our analyses come from surveys in two German cities, Mainz and Hanover, and two Swiss cities, Bern and Zurich. Mainz is located in the middle of Germany near Frankfurt and has about 220,000 inhabitants. Hanover is more in the north of Germany and has about 530,000 inhabitants. Bern is the capital of Switzerland, and about 130,000 people live there. Zurich is the biggest city of Switzerland and has about 430,000 inhabitants. Pragmatic considerations were important for the selection of these cities. Members of our research group were affiliated with the universities in three of the four cites, and this proved to be helpful both for the survey sampling and for access to the “objective” noise exposure data.

With the exception of a few local adaptations, the surveys in the four cities were strictly comparable in terms of research design (sampling procedure, etc.) and question program. The surveys were carried out as mail questionnaires and were conducted between October 2016 and March 2017. They were based on random samples of the adult population (aged 18–70) in the four cities. The addresses of the random samples, which specified individual persons, came from the official population registers managed and maintained by the city administrations. Because we had the exact addresses of our respondents, we were additionally able to locate the spatial coordinates of their places of residence. These coordinates enabled us to match administrative noise data to the survey data (Section 3.2).

Subjects selected for participation in the study were approached following the tailored design method of Dillman et al. [41]. That is, they received a first invitation to participate in the survey, a postcard after one week, a second invitation after three weeks, and a third invitation after seven weeks. It is important to note that the surveys were not introduced as an environmental survey, but as a survey titled “Housing and Living in [City Name]”. Starting with 4000 addresses in each city, the surveys yielded a response rate of 45.2 per cent in Mainz, 35.9 per cent in Hanover, 55.2 per cent in Bern, and 48.4 per cent in Zurich (standard RR2 for postal surveys to specifically named persons [42]). In total, 7540 respondents participated in the survey (for further details of the study, including issues of sample selectivity, see [43]).

For our analyses, we excluded some cases from the beginning because the answers showed considerable inconsistencies and/or gave hints that the data would be unreliable. Furthermore, we use only complete cases—that is, cases with valid values for all variables. This reduces the number of cases to 5301.

### 3.2. Variables and Their Operationalization

In Figure 1, we have five “boxes” of variables for which we need empirical measures: noise annoyance; socioeconomic variables; noise exposure; housing attributes; and environmental susceptibility. A further group not shown in Figure 1 is a set of covariates that serve as statistical controls. In this section, we merely describe the measurement of these variables—without descriptive statistics, which will be given in Section 3.3.

*Noise annoyance.* Our crucial dependent variable is annoyance resulting from residential road traffic noise. For its measurement we used the standard 11-point scale, ranging from “0 = not annoyed at all” to “10 = very much annoyed” [44]. However, we modified the standard item. We did not ask respondents to think about the last 12 months when they were at home, but—without specifying a timeframe—to think about their situation at home under four different conditions. Our question wording was as follows: “When you are at home in your dwelling, how strongly do you feel annoyed by road traffic noise, (1) during the day when the dwelling’s windows are open, (2) during the day when windows are closed, (3) during the night when windows are open, and (4) during the night when windows are closed?” Adding up the answers for the four constellations and dividing the sum by 4 yields our dependent variable “noise annoyance,” with a range from 0 to 10.

*Socioeconomic variables.* Based on the topic of interest in this article, socioeconomic characteristics are the most important independent variables. As already described in Section 2.2, we refer to migration background, education, and income for their measurement. Depending on the country of origin, a respondent is assigned a migration status independently of citizenship. We distinguish (a) no migration background, (b) European and other Western countries (North America, Australia), and (c) Africa, Asia, and South America. Although group (c) is very heterogeneous, shortages of socioeconomic resources as well as difficulties of social integration can be assumed to be more pronounced in this group. Education is measured by years of schooling typically needed to achieve a specific educational level. A household’s income situation is captured by the monthly net equivalent household income, using the modified OECD scale. To make incomes comparable between Germany and Switzerland, we converted Swiss Francs into Euros and adjusted the income according to purchasing power parity (PPP). This means that (monthly net equivalent) “household income” is measured in PPP-adjusted Euros.

*Noise exposure.* The noise exposure data were not gleaned from the survey, but from external sources, i.e., from administrative noise registers in the four cities. The addresses of our respondents denoted their exact place of residence. We first determined the spatial coordinates for these locations. Based on these coordinates, fine-grained data on local road traffic noise were merged with the survey data. Fine-grained means the data focus directly on the building where the respondents lived. To capture the level of noise exposure we refer to the day-evening-night level (Lden). This measures noise exposure in decibels dB, gives a weighted 24 h average, and applies the usual penalties for evening and nighttime noise [45]. Section A.1 provides additional information about our noise exposure data.

*Housing attributes.* In Section 2.2, we introduced the four housing attributes “dwelling size,” “bedroom facing the street,” “window quality,” and “dwelling with outdoor garden”. Dwelling size is measured in m^2^ (divided by 10). Whether or not the respondent’s bedroom faces the street is a dummy variable. The respondents assessed the quality of the windows of their dwelling on a 5-point scale, ranging from “1 = very bad” to “5 = very good”. Dwelling with outdoor garden registers the binary information on whether the respondent’s residence has a private garden.

*Environmental susceptibility*. Environmental concern and noise sensitivity are the two variables in this group. The measurement of environmental concern refers to six items of the environmental concern scale of Diekmann and Preisendörfer [46]. Noise sensitivity was measured by an additive index of five items, adapted from Weinstein’s noise sensitivity scale [47,48]). Details on the measurement of environmental concern and noise sensitivity are summarized in Section A.2. 

*Control variables.* As statistical controls, we use five variables: gender; age; labor force participation; household size; and city. Gender is included as a dummy with “1 = woman”. Age is measured in years (divided by 10). Labor force participation is another dummy variable with “1 = currently active in the labor market”. Household size registers the number of persons in the household. With Mainz as reference category, “city” captures the urban context in the form of three further dummy variables. 

### 3.3. Statistical Procedures and Descriptive Statistics

To examine whether and how socioeconomic background affects annoyance resulting from traffic noise in the neighborhood, we follow a standard three-step procedure of mediation analysis [49]. This procedure recommends three different regression models that allow a separation of the total, direct, and indirect effects of our socioeconomic variables: (1) a regression that includes the socioeconomic variables, but excludes the supposed mediators to get the total effects; (2) a regression that includes both the socioeconomic variables and the supposed mediators to get the direct effects; and (3) regressions of the mediators on the socioeconomic variables to get the first step of the indirect effects.

The descriptive statistics of the variables relevant for these regressions are given in Table 1. Concerning the socioeconomic variables, the table shows that 9% of the respondents have a Western and 16% a non-Western migration background. The mean of education is 15.2 years. The average PPP-adjusted household income is 2930 Euros.

**Table 1 ijerph-19-14984-t001:** Variables and their descriptive statistics.

Variable	Obs.	Mean	Std. Dev.	Min.	Max.
Noise annoyance	5301	2.26	2.29	0	10
Migration background					
No	3997	0.75	0.43	0	1
Yes, European/Western	445	0.09	0.28	0	1
Yes, non-Western	859	0.16	0.37	0	1
Education in years	5301	15.2	2.73	8	18
Household income/1000	5301	2.93	1.47	0.20	10.00
Road traffic noise Lden	5301	52.96	7.29	32.17	81.09
Dwelling size in m^2^/10	5301	9.36	4.22	0.80	30.00
Bedroom facing the street	5301	0.52	0.50	0	1
Window quality	5301	3.71	1.06	1	5
Dwelling with outdoor garden	5301	0.48	0.5	0	1
Environmental concern	5301	3.53	0.78	1	5
Noise sensitivity	5301	3.16	0.87	1	5
Woman	5301	0.54	0.50	0	1
Age in years/10	5301	4.25	1.36	1.80	7.00
Labor force participation	5301	0.75	0.44	0	1
Household size	5301	2.46	1.19	1	8
City					
Mainz	1219	0.23	0.42	0	1
Hanover	906	0.17	0.38	0	1
Bern	1686	0.32	0.47	0	1
Zurich	1490	0.28	0.45	0	1

Although noise annoyance, our final dependent variable, is not normally distributed (mean = 2.3 and median = 1.5) and therefore OLS regressions do not fit exactly, we decided in favor of OLS models—instead of binary logistic regressions (with % highly annoyed as dependent variable). OLS models have the advantage that they exploit the data of the 11-point scale more fully than 0/1 logistic regressions. To account for the city contexts, all the models incorporate the city dummies Hanover, Bern, Zurich as fixed effects (Mainz serves as reference). Furthermore, all models use gender, age, labor force participation, and household size as statistical controls.

## 4. Empirical Results

### 4.1. Total Effects of Socioeconomic Variables

Models 1a to 1c in Table 2—the so-called reduced-form regressions that leave out the endogenous mediator variables—are appropriate to capture the total effects of our socioeconomic variables (i.e., their direct and indirect effects together). The table displays unstandardized regression coefficients and—in parentheses—their absolute t-values. Model 1a shows significant positive total effects of the two migration background dummies. This means that (as expected) respondents with a migration background complain more often about road traffic noise in their neighborhood. According to Model 1b, education tends to have a negative total effect, but the effect is not significant. Household income, however, yields a highly significant negative total effect on annoyance, and this is in line with expectations from environmental inequality research.

**Table 2 ijerph-19-14984-t002:** Factors affecting annoyance due to road traffic noise (OLS regressions).

	Model 1a	Model 1b	Model 1c	Model 2
Migration background				
European/Western	0.25 * (2.10)			0.08 (0.76)
Non-Western	0.26 ** (3.03)			0.03 (0.34)
Education in years		−0.02 (1.55)		0.01 (0.15)
Household income/1000			−0.17 *** (7.08)	−0.06 * (2.36)
Road traffic noise Lden				0.12 *** (33.41)
Dwelling size in m^2^/10				−0.01 (0.81)
Bedroom facing the street				0.82 *** (15.35)
Window quality				−0.43 *** (17.16)
Dwelling with outdoor garden				−0.26 *** (4.55)
Environmental concern				0.19 *** (5.54)
Noise sensitivity				0.50 *** (16.15)
Woman	−0.03 (0.45)	−0.03 (0.51)	−0.07 (1.14)	−0.10 (1.88)
Age in years/10	−0.14 *** (5.96)	−0.14 *** (6.07)	−0.12 *** (5.22)	0.01 (0.17)
Labor force participation	−0.02 (0.23)	0.01 (0.05)	0.13 (1.72)	0.09 (1.48)
Household size	−0.02 (0.71)	−0.01 (0.45)	−0.03 (1.33)	0.03 (1.02)
City				
Hanover	−0.13 (1.27)	−0.12 (1.22)	−0.16 (1.55)	−0.46 *** (5.44)
Bern	−0.33 *** (3.79)	−0.31 *** (3.55)	−0.19 * (2.16)	0.01 (0.14)
Zurich	−0.02 (0.27)	0.04 (0.41)	0.20 * (2.17)	0.18 * (2.23)
Constant	2.99 *** (19.60)	3.30 *** (13.84)	3.32 *** (21.02)	−5.11 *** (15.45)
Adj. R^2^	1.1%	1.0%	1.8%	31.8%
No. of cases	5301	5301	5301	5301

Notes: Unstandardized regression coefficients with absolute t-values in parentheses. * *p* < 0.05, ** *p* < 0.01, *** *p* < 0.001.

Nevertheless, the fit values (adj. R^2^) of Models 1a–1c are very low. Thus, when it comes to annoyance caused by traffic noise, social background variables play a certain role, but it is evident that they are not dominant predictors.

### 4.2. Direct Effects of Socioeconomic Variables

The effect pattern of the socioeconomic variables clearly changes when Model 2 in Table 2 additionally takes the supposed mediators (noise exposure, housing attributes, and environmental susceptibility) into account and thus shifts the analysis to the direct effects of the socioeconomic variables. Both migration dummies are no longer significant. Supplementary analyses (not shown here) reveal that it is mainly the inclusion of noise exposure and environmental concern that is responsible for the dropping away of the migration background effects. Whereas the total education effect pointed in a negative direction, there is definitely no direct effect of education on noise annoyance. Supplementary analyses (again not shown here, but see Table 3 below) suggest that if education plays a role, it is mainly via its influence on environmental concern and on noise sensitivity. Contrary to our prediction, the direct income effect remains statistically significant in Model 2. However, compared to the total income effect, the direct effect is much lower. The total income effect is nearly three times as strong as the direct effect (Model 1c versus Model 2). This implies that the indirect income effects contribute more to the reduced noise annoyance of high-income households than the direct effect. It further implies that empirical studies that restrict their interest to the direct income effects (as is the case with most studies in noise research) underestimate the importance of income and financial resources for subjective annoyance due to road traffic noise.

Gauged by the size of the t-values of the regression coefficients in Model 2, noise exposure is the most important influence factor on noise annoyance, followed by window quality, noise sensitivity, and whether or not the bedroom faces the street. The findings with respect to noise exposure and noise sensitivity correspond to prior studies in noise research (Section 2.1). The effect of the quality of the windows is remarkably strong, and this means in practice that high-quality soundproofed windows are a very effective way to reduce residential noise annoyance. The significant positive effect of environmental concern on noise annoyance is a finding that up until now has not been a prominent topic in noise research. Contrary to our expectations, dwelling size shows no direct effect in Model 2. With adj. R^2^ = 31.8%, Model 2 fits the data much better than Models 1a–1c. Compared to the direct effects of noise exposure, noise sensitivity and the two housing attributes of window quality and location of the bedroom, the direct effect of income is small and much weaker.

Taken together, the regression coefficients in Model 2 confirm the hypothesis of missing or at least only weak direct effects of socioeconomic variables on noise annoyance. Although this more or less corresponds to “the prevailing view” in noise research (see again, Section 2.1), we can claim for our study that our measurement of social background characteristics has been more refined than that used in other noise studies. Additionally, based on this refined measurement, we have found that if there are direct effects of socioeconomic variables, they mainly hinge on income and financial resources shielding against noise annoyance. 

### 4.3. Indirect Effects of Socioeconomic Variables

This “shielding capacity” of financial resources becomes more evident when we look at the indirect pathways theoretically suggested and explained above. Figure 1 specified seven potential mediator variables: noise exposure, four housing attributes, and two environmental susceptibility variables. Using them as dependent variables, Table 3 presents the results of OLS regressions; each includes the socioeconomic variables and our set of control variables. According to Model 2 in Table 2, six of the seven supposed mediators show significant direct effects on noise annoyance, and we therefore focus on these mediators. The exception is the housing attribute “dwelling size” that—contrary to our expectations—does not (at least not directly and independent of the other housing attributes) contribute to a significant reduction of noise annoyance.

**Table 3 ijerph-19-14984-t003:** Factors affecting noise exposure, housing attributes, and environmental susceptibility (OLS regressions).

	Road Traffic Noise Lden	Dwelling Size	Bedroom Facing the Street	Window Quality	Dwelling with Outdoor Garden	Environmental Concern	Noise Sensitivity
Migration background							
European/Western	0.77 * (2.08)	−0.71 *** (4.21)	0.04 (1.49)	0.01 (0.18)	−0.05 * (2.16)	−0.08 * (2.00)	0.09 * (2.06)
Non-Western	1.00 *** (3.60)	−1.11 *** (8.82)	0.06 ** (3.22)	−0.14 *** (3.37)	−0.04 * (2.40)	−0.31 *** (10.58)	−0.07 * (2.05)
Education in years	−0.01 (0.37)	0.03 (1.85)	−0.01 (1.48)	0.01 (1.65)	0.01 *** (3.71)	0.04 *** (9.36)	0.03 *** (6.06)
Household income/1000	−0.29 *** (3.54)	1.11 *** (30.44)	−0.03 *** (4.68)	0.10 *** (8.77)	0.03 *** (5.42)	−0.07 *** (7.77)	0.02 * (2.24)
Woman	−0.32 (1.63)	0.32 *** (3.62)	−0.01 (0.13)	0.03 (0.97)	0.04 ** (2.99)	0.20 *** (9.47)	0.12 *** (4.82)
Age in years/10	−0.61 *** (8.18)	0.70 *** (20.81)	−0.01 * (2.57)	0.07 *** (6.76)	0.07 *** (13.41)	−0.02 * (2.55)	0.04 *** (4.20)
Labor force participation	−0.06 (0.27)	−0.46 *** (4.24)	−0.01 (0.42)	−0.04 (1.10)	−0.01 (0.88)	−0.01 (0.07)	0.03 (1.01)
Household size	−0.31 *** (3.66)	1.93 *** (51.00)	0.02 *** (4.24)	0.03 * (2.51)	0.11 *** (19.08)	0.03 ** (2.89)	−0.04 *** (3.64)
City							
Hanover	2.44 *** (7.78)	0.23 (1.60)	−0.04 (1.92)	−0.04 (0.79)	0.09 *** (4.24)	0.02 (0.60)	0.07 (1.73)
Bern	−0.61 * (2.19)	−0.80 *** (6.38)	−0.01 (0.27)	0.07 (1.75)	0.06 *** (3.49)	0.14 *** (4.76)	−0.26 *** (7.66)
Zurich	0.47 (1.59)	−1.14 *** (8.53)	−0.06 ** (2.75)	0.02 (0.57)	−0.12 *** (6.38)	0.10 ** (3.25)	−0.19 *** (5.30)
Constant	57.00 *** (73.81)	−1.18 *** (3.38)	0.68 *** (12.60)	2.88 *** (25.58)	−0.29 *** (5.76)	3.01 *** (37.14)	2.63 *** (28.43)
Adj. R^2^	4.1%	41.7%	1.9%	3.8%	11.3%	6.6%	4.0%
No. of cases	5301	5301	5301	5301	5301	5301	5301

Notes: Unstandardized regression coefficients with absolute t-values in parentheses. * *p* < 0.05, ** *p* < 0.01, *** *p* < 0.001.

With respect to the “noise exposure path”—the indirect pathway rooted in environmental inequality research—Table 3 reveals that respondents with a migration background are exposed to higher noise levels at their place of residence, while respondents with a higher income are exposed to lower noise levels. Education, however, is not significant. This is in line with our hypotheses.

Concerning the “housing attributes path”—inspired by the environmental shielding hypothesis [31] —we also find a confirmation of our expectations. Respondents with a migration background, and particularly those with an origin in a non-Western country, live more often in dwellings with a bedroom facing the street, reside less often in dwellings with high-quality windows, and less often have dwellings with a garden. The opposite is true for those with a high income. The income effects tend to be stronger than the effects of migration background. Education shows in the context of the housing attributes only one significant effect: respondents with more years of schooling live more often in dwellings with a garden.

The third indirect pathway suggested in Figure 1, the “environmental susceptibility path,” argues in favor of a mechanism that runs counter to the overall tendency for privileged social groups to experience lower noise annoyance. For education, the results in Table 3 clearly support the conjecture that highly educated respondents are both environmentally more concerned and more sensitive toward noise. The effects of education stay significant when we additionally control for noise exposure in the regression equations of environmental concern and noise sensitivity (not shown in Table 3). Less clear-cut are the effects of income and migration background. Income yields the expected positive effect on noise sensitivity, but its effect on environmental concern is negative. As predicted, respondents with a migration background are environmentally less concerned than those without such a background. With respect to noise sensitivity, the regressions in Table 3 show a negative effect for respondents with a non-Western, but a positive effect for those with a Western migration background. 

To check the robustness of our results, Section A.3 discusses further empirical analyses, including an estimation of all regressions in the form of a structural equation model (SEM).

## 5. Discussion and Conclusions

To understand whether and how social background variables relate to individual environmental risks, it seems reasonable to begin with two basic insights. (1) Objective exposure to environmental risks should be separated from their subjective perceptions and assessments. These two sides often do not go hand in hand, and—given the same level of exposure—education, income, and other socioeconomic variables may induce different subjective reactions. (2) The effects of socioeconomic variables on the subjective perceptions and evaluations of environmental risks tend to be underestimated, because they are not direct, but predominantly indirect. They are mediated by other variables, implying indirect paths, which have to be taken into account in adequate empirical appraisals.

We have demonstrated the validity of these two basic and more general propositions in an empirical application to noise annoyance caused by residential road traffic. From ample noise research, we know that besides objective noise exposure several other factors influence subjective noise annoyance. Our results clearly support this insight. However, whereas mainstream noise research regularly states that education, income, social status, etc. do not have significant effects on noise annoyance, we find that they do have such effects, albeit mainly indirect. Based on our theoretical model and our empirical results, three indirect paths and mechanisms deserve special attention: a noise exposure path; a housing attributes path; and an environmental susceptibility path.

Socioeconomic resources create opportunities for individuals to choose a place of residence with lower noise exposure, and this usually also reduces noise annoyance. When there is road traffic noise in the neighborhood, individuals and households with more socioeconomic resources can shield themselves better against this noise via more favorable housing conditions. They have larger dwellings, less often a bedroom facing the street, more often high-quality soundproofed windows, and more often a backyard garden. Our analyses demonstrate that high-quality windows in particular effectively reduce indoor noise and thus noise annoyance. These structural advantages notwithstanding, if road traffic noise characterizes a neighborhood, privileged households tend to react with more feelings of anger than less privileged ones. They have and “can afford” a higher noise sensitivity and are environmentally more concerned, and this stimulates—at the same level of noise exposure—more annoyance. The finding of an increased environmental susceptibility on the part of privileged people is well in line with the assumption of local environmental quality as a “luxury good” [50]. 

Looking at the relative strength of migration background, education, and income as separate aspects of the socioeconomic standing, our results suggest that migration background and income mainly unfold along the noise exposure and housing attributes paths. Thereby, income tends to be more important than migration background. Education, on the other hand, is most relevant to the environmental susceptibility path.

Like other studies, our study has limitations and weaknesses. All our findings pertain to annoyance due to urban road traffic noise. It would be desirable to apply our model with its three indirect pathways to other local environmental risks. If the results also hold for the subjective assessments of other risks, this could strengthen the conclusion that the effects of unequal socioeconomic resources are more profound than is often presumed in noise research. 

Our study pertains to four European cities and thus has a local restriction. At best, the selected cities can be seen as more or less typical European cities, but surely do not represent cities in less developed countries. Of course, it would be preferable to investigate additional cities—cities in other parts of the world and cities with more pronounced social inequalities. 

The noise exposure path rests on the assumption of selective migration processes in reaction to traffic noise. Our cross-sectional data do not allow us to test this assumption, but our literature review revealed that the empirical validity of selective move-out and move-in processes, induced by local environmental bads, is far from trivial. 

The relationship between outdoor road traffic noise and subjective noise annoyance in the dwelling is mediated by the indoor noise level, which certainly differs from the outdoor noise level. Particularly with respect to the housing attributes path, it would be helpful to know this level of indoor noise, but we did not have data on indoor noise exposure. Furthermore, it would be useful to take other characteristics of the buildings (e.g., whether they have façade insulation) and the dwellings (e.g., the internal arrangement of the rooms) into account to gain a better understanding of social differences in private noise prevention strategies.

Our finding that people with lower education and lower income and with a non-Western migration background have a lower noise sensitivity should not be misunderstood. It does not necessarily mean that less privileged people personally suffer less from a given level of noise and that the negative health consequences of noise are less serious. It is well known that the objective noise level is detrimental to health even when people seemingly adapt to it. Privileged social groups are more eager to voice protests against noise and more engaged in active opposition, and this may actually spill over into stronger feelings of annoyance. Consequently, it seems to be a challenging research topic to elaborate the details of the interrelationships between socioeconomic variables, noise sensitivity, and other subjective reactions to noise.

## Figures and Tables

**Figure 1 ijerph-19-14984-f001:**
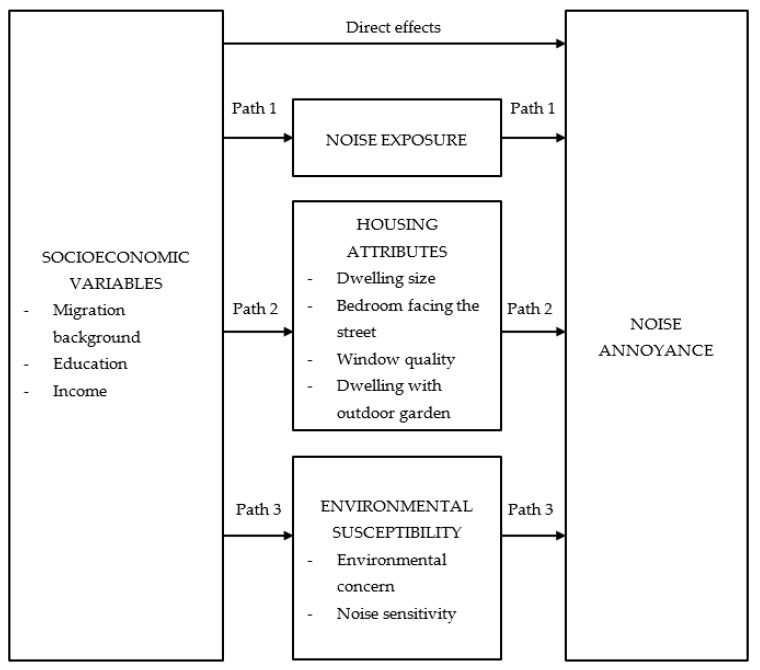
Potential direct and indirect effects of socioeconomic variables on noise annoyance.

## Data Availability

The data are accessible upon request from the authors (see also https://search.gesis.org/research_data/SDN-10.7802-1993, accessed on 10 October 2022).

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
