# Peer review of "Pathways to Environmental Inequality: How Urban Traffic Noise Annoyance Varies across Socioeconomic Subgroups"

_ijerph, 2022, doi:10.3390/ijerph192214984_

Round 1
Reviewer 1 Report
Paper presented is well structured and interesting research questions are discussed in this present frame.
More specically,
1. in & introduction and background and hypotheses :
Authors might also say that many research have tried to find a dose-response model where annoyance can be quantified during the 80th.
Even though, the short Leq and the dB(A) are also "tools" to bridge quantitative evaluation and qualitative evaluation.
It's a good introduction, in my opinion, to overpass these problematics. and bringing the debate on another level, as the authors say : "We intend to show in this article that both sides, environmental inequality research 63 on the one hand and noise research on the other, could benefit from being brought into a 64 closer relationship." (line 62-63)
In the Socioeconomic variables, we wonder if "property" could be a variable and if it measurable (how can you get these infos) - it not always linked to the incomes.
The idea of "noise exposure path" is also very interesting and eases to avoid any dead end in the correlation process. Noise annoyance and noise exposure are dynamic notions (changing with the time - months, years, decades maybe...)
The "“environmental shielding hypothesis”" also discussed in line 232-251 also discussed in line 232-251 are also very important. Maybe the authors should detail more the "real" efficiency (or not) of such "devices - hypothsesis"
Then the 3rd path of analysis, environmental susceptibility path, is really rich and might express the population in the survey ? Are they any historic evaluation of this 3rd axe during the 90s, 00s, 2010s ?
3. Data, Variables and Methods
Selection of the 4 cities understandable and logic to possibility to use already produced studies (regarding as far as I know, Noise Strategic maps and Noise action plans ? - European Directive 2002/EU/49 on the environmental noise.
Methdologies of research are very clear, tables of results clearly presented and perfectly discussed.
Maybe some cartographic representation of the results might have be very interesting in order to include "spatial - architectural paths" in the general discussion.
Bibliography is up to date and well presented in the text.
For all these reasons, we strongly recommend the publication of the paper.
Author Response
Many thanks for your constructive and helpful recommendations. We tried to follow your recommendations in the following way:
We now mention previous research on the “dose-response model” in Section 2.1, and we added two further literature references (Guski et al., 2017; Gjestland, 2020), both published in IJERPH.
In a new footnote at the end of Section 3.2, we concede that “property”, i.e. homeownership may play a role. However, homeownership significantly correlates with income, which is one of our essential variables of interest. We argue that – based on the methodological principle of parsimony – we restricted our analyses to a core set of statistical controls.
Concerning the “efficiency” of the environmental shielding via housing attributes, we now explicitly mention the remarkably strong effects of high-quality soundproofed windows on noise annoyance in both the results Section 4.2 and the discussion Section 5.
Concerning the environmental susceptibility path, our findings pertain only to the four cities investigated in our study. This means that the findings “express the population in the survey”. Nevertheless, our samples are high-quality random samples of the four cities. In Section 2.2, where we discuss the environmental susceptibility path, we list some references, which also deal with this path in diverse national contexts. Regrettably, we do not have historic evaluations of “this 3rd axe during the 90s, 00s, 2010s”. As far as we know, our results are relatively innovative with respect to this path.
Reviewer 2 Report
Dear Editor, dear authors;
The European Green Deal has been presented by the European Commission with the objective of nothing more and nothing less than achieving zero pollution. EU policies include environmental noise in this ambition. Environmental data managed by the European Environment Agency (EEA) reveals that a substantial proportion of the urban population in the EU is still exposed to noise levels that exceed the reference values proposed by the World Health Organization (WHO). In this context, the authors of the manuscript IJERPH-1993033, entitled: Pathways to Environmental Inequality: How Urban Traffic Noise Annoyance Varies Across Socioeconomic Subgroups, the first author is: Peter Preisendörfer, focus their study on trying to show that socioeconomic conditions and environments with low noise quality are linked in Europe. Therefore, it can be affirmed that the subject of study is current since there is a growing political and scientific interest in Europe in this type of research. (For example (i) European Environment Agency. Unequal exposure and unequal impacts: social vulnerability to air pollution, noise and extreme temperatures in Europe, EEA Report No 22/2018. Copenhagen: EEA; 2019. (ii) WHO Regional Office for Europe. Environmental health inequalities in Europe. Second assessment report. Copenhagen: WHO Regional Office for Europe; 2019).
On first reading, the manuscript reveals a narrative (I would almost say vibrant) quality that makes it very easy to read. I also believe that behind the manuscript there is a careful selection of the content of the supplementary material and its writing, which is appreciated. It can be said that the extension of the article is long, but I would not dare to remove anything. There are some repetitions, but they are not very important. I want to say that I enjoyed the first reading of the article (and this is not usually said in a scientific review).
The title is correct and together with the abstract they give a clear idea of the nature of the work. The research questions and working hypotheses are clear and achievable with the proposed methodology. The introduction (including sections 1 and 2) to the hot topics is very clear. The explanations are easy to follow. The scientific environment in which the research is carried out is correctly introduced. In this work, the socioeconomic status (through the 3 measurable variables: migratory group, education, and income) of each citizen is the most important independent variable. Also included are 3 variables (called “mediators”: 1) exposure to noise through the results of noise maps, 2) some attributes of the dwelling, and 3) personal level of environmental awareness). All of this is as suggested in Figure 1. Together with the dependent variable, the authors have obtained (using the proposed regression models): the total, direct and indirect effects of the socioeconomic variables. I honestly do not think it is necessary to explain the statistical tools used beyond what is included in section 3.3. and Appendix 3.
Two aspects of the scope of work are worth noting.
- The noise source is traffic noise. Probably because this type of noise affects all cities and a large proportion of their population. In this way, it is easier for the authors to obtain 5,301 responses in four cities (2 countries and more than one million inhabitants).
- Noise receivers are the people in the home. These data on the population exposed to noise on the facade of residential buildings are available due to the obligation to calculate them in Europe in cities with more than 100,000 inhabitants (END). And in Switzerland, I know that similar work has been going on for many years.
There are many more positive aspects to underline, but I think it is best to get straight to the point:
(i) In my humble opinion it is a great article and I recommend its publication in the Int. J. Environ. Res. Public Health. I dare to anticipate that many colleagues will read this article.
And (ii) although the article is publishable in its current state, I think it is necessary to point out a series of aspects to improve. I think it's worth discussing some doubts before the final publication. I would like to hear what the authors have to say about my suggestions. These minor corrections are summarized in the following:
· Section 1. I miss (at least) a mention of the systematic review written by Stefanie Dreger, Steffen Andreas Schüle, Lisa Karla Hilz, and Gabriele Bolte, entitled: Social Inequalities in Environmental Noise Exposure: A Review of Evidence in the WHO European Region, published precisely in this journal on March 20, 2019.
· Subsection 2.2. (and others) Regarding migration background. Do the authors think that an urban Japanese can be compared with a Vietnamese who come from the countryside? Do you think that the rural or urban origin factor can be more relevant than the continent or country origin to explain noise annoyance? Furthermore, do the authors think that the cultural factor may be relevant in the interpretation of noise annoyance?
· Subsection 2.2. (and others) Perhaps I have overlooked the explanation of whether the houses reviewed are directly comparable in terms of their design. I am referring to the typology of buildings (large condominiums, multi-story buildings, single-family homes, etc.) that have been the subject of the surveys.
· Although correct, Figure 1 looks a bit poor. Perhaps it could be improved, for example including information on the analysis paths, and the control variables.
· In tables 2, 3, and 4 the city of Mainz has disappeared.
· Line 351 and Appendix 1. A piece of advice. Please don't use "dB(A)" when it comes to noise indicators like Lden. Please remove “dB(A)” and use “dB” instead.
· Section 5. The formal structure of the paper (in general) is correct but section 5 Discussion and Conclusions. Actually, it seems more like a summary with reflections on the work, than really a discussion. So my opinion is that section 5 should be improved. How? I will try to give a possible way (always trying to be constructive).
1) The conclusions should be explicitly contained in an isolated “conclusions section 5”.
2) Perhaps in this article, it would be more appropriate to create a new Section “4. Results and Discussion”, where the authors can include (as a 4.4.) a short discussion providing more insights, rearranging some content of the previous subsections 4.1, 4.2 and 4.3., and perhaps comparing the results with other studies, and interpreting the new findings.
3) The “limitations of the study” in subsection 4.5.
In any case, I am sure that the authors will appreciate my concern and make the right decision.
· Appendix 1 explains very clearly how the exposure data has been extracted and processed. A good job regarding exposure data harmonization. But I still have a doubt concerning the road traffic noise calculation models that have been used in the 4 cities (sonRoad, CNOSSOS, RL-90,…?) and if their results are directly comparable. In general, the question is whether the methodology for making the maps on both sides of the border is comparable.
Author Response
Many thanks for your constructive and helpful recommendations. We tried to follow your recommendations in the following way:
It really was a serious flaw that we missed the review article of Dreger et al. (2019), published in IJERPH. Sorry, we simply were not aware of this important article. Now, in our revised version of the manuscript, we cite this article three times. Again, thanks for this hint.
Of course, you are right that our migration background dummy “non-Western background” combines a highly diverse group of respondents. We now explicitly mention the great heterogeneity of this group in Section 3.2. It may be that rural versus urban origin of the respondent is also a relevant factor, but we do not have this information in our data. Nevertheless, in everyday life, the native population tends to judge people in such crude categories as Western versus non-Western migration background, and these judgments influence everyday behavior (e.g., discrimination in the housing market, and thus effects on the quality of housing of our respondents).
Our data does not include information on the typology of the buildings where the respondents live. We directly asked for those features of the buildings (window quality, size of the dwelling, etc.) that are seen as important in the noise annoyance literature.
We concede that Figure 1 may look “a bit poor”. We now explicitly included the direct effects and the three indirect paths (path 1, 2, 3). Generally, we are willing to concede that the causal structure may be more complex than the simple diagram in Figure 1. It may be, for example, that noise sensitivity is negatively associated with noise exposure because noise-sensitive persons tend to choose quieter dwellings. On the other hand, a higher level of noise exposure may have the consequence that persons become more sensitive toward noise – resulting in a positive association between noise sensitivity and noise exposure. Our intention was to focus here on those mechanisms that are theoretically most plausible.
In Tables 2-4 the city of Mainz serves as reference category (and therefore “disappeared”), whereas Hanover, Bern, and Zurich are three city dummies. We explicitly mention this now at the end of Section 3.3.
We removed dB(A) and use now dB. Thanks for this valuable hint.
We did not follow your recommendation to rearrange Section 5. We know that there are different traditions in different scientific disciplines on how the final section of a paper should look like. Your proposal follows the tradition in the medical and natural sciences. Our structuring of Section 5 follows the tradition in the social sciences. We hope you can accept this decision. A restructuring of Section 5 (together with Section 4) finally would not change the baseline story of our article.
At the end of Appendix 1, we added with respect to the comparability of the German and the Swiss noise data the following remark: “Given the methodological differences in the measurement of residential noise exposure in the two German and the two Swiss cities, direct comparisons to the German and Swiss administrative noise data should be handled with reasonable care.”
Reviewer 3 Report
This paper is very well written and easy to read. I would like the following minor changes.
Point 1: The characters "garden" in Figure 1 are not displayed. Point 2: Please explain somewhere what the numbers in Table 2 mean. Point 3: Regarding Table 3, it is difficult to read because the position of “European/Western” is shifted down by one line. Also, it would be easier to understand if “European/Western” is right-aligned like “Non-Western”. Point 4: Did you check the correlation coefficient between “Non-Western”, “Household income/1000” and “Age in years/10”? High correlation coefficients between these variables can lead to multicollinearity and unstable parameter effects and significance values if they are included in the model at the same time. It's better to check the correlation coefficients and describe that there doesn't seem to be a big problem. Also, if the correlation is very high, one of the highly correlated variables should be removed from the model and reanalyzed. Point 5: Path analysis or SEM(Structural Equation Modeling) are commonly used to analyze the direct and indirect effects of specific parameters. Why didn't you choose that method? If you have any reason, please write it down. If you plan to work on it as the next step, I think you should write about it.
Author Response
Many thanks for your constructive and helpful recommendations. We tried to follow your recommendations for minor changes in the following way:
Point 1: The characters "garden" in Figure 1 are not displayed.
The transfer of our original manuscript to the IJERPH-format caused this problem. We have changed Figure 1 to eliminate the problem.
Point 2: Please explain somewhere what the numbers in Table 2 mean.
At the beginning of Table 2 we now explicitly say that table displays unstandardized regression coefficients and – in parentheses - their absolute t-values.
Point 3: Regarding Table 3, it is difficult to read because the position of “European/Western” is shifted down by one line. Also, it would be easier to understand if “European/Western” is right-aligned like “Non-Western”.
The transfer of our original manuscript to the IJERPH-format caused this problem. We have changed Table 3 to eliminate the problem.
Point 4: Did you check the correlation coefficient between “Non-Western”, “Household income/1000” and “Age in years/10”? High correlation coefficients between these variables can lead to multicollinearity and unstable parameter effects and significance values if they are included in the model at the same time. It's better to check the correlation coefficients and describe that there doesn't seem to be a big problem. Also, if the correlation is very high, one of the highly correlated variables should be removed from the model and reanalyzed.
We checked the correlation coefficients of our set of independent variables (used in the models of Table 2 and 3). None of the correlations is higher than 0.4, and thus multicollinearity should not be a problem. The VIF values of all models are moderate.
Point 5: Path analysis or SEM (Structural Equation Modeling) are commonly used to analyze the direct and indirect effects of specific parameters. Why didn't you choose that method? If you have any reason, please write it down. If you plan to work on it as the next step, I think you should write about it.
Actually, as a robustness check, we present our regression results also as SEM, see Appendix 3 of our Supplementary Material (Table A1). Our results are stable and robust when we estimate a structural equation model considering all paths simultaneously. We finally did not choose the SEM method in the main text of our article for two reasons: (1) In some scientific disciplines, SEMs are not common, and IJERPH is an interdisciplinary journal. (2) We prefer models that are based on a minimum of assumptions. Separate models are more robust against the violation of statistical assumptions than a model that estimates all coefficients simultaneously in a single step.